# Exploratory study reveals far reaching systemic and cellular effects of verapamil treatment in subjects with type 1 diabetes

Guanlan Xu[1,2], Tiffany D. Grimes[1,2], Truman B. Grayson[1,2], Junqin Chen[1,2], Lance A. Thielen[1,2], Hubert M. Tse [1,3], Peng Li [1,4], Matt Kanke[5], Tai-Tu Lin[6], Athena A. Schepmoes[6], Adam C. Swensen[6], Vladislav A. Petyuk [6], Fernando Ovalle [1,2], Praveen Sethupathy[5], Wei-Jun Qian [6] & Anath Shalev [1,2✉]

Currently, no oral medications are available for type 1 diabetes (T1D). While our recent randomized placebo-controlled T1D trial revealed that oral verapamil had short-term beneficial effects, their duration and underlying mechanisms remained elusive. Now, our global T1D serum proteomics analysis identified chromogranin A (CHGA), a T1D-autoantigen, as the top protein altered by verapamil and as a potential therapeutic marker and revealed that verapamil normalizes serum CHGA levels and reverses T1D-induced elevations in circulating proinflammatory T-follicular-helper cell markers. RNA-sequencing further confirmed that verapamil regulates the thioredoxin system and promotes an anti-oxidative, anti-apoptotic and immunomodulatory gene expression profile in human islets. Moreover, continuous use of oral verapamil delayed T1D progression, promoted endogenous beta-cell function and lowered insulin requirements and serum CHGA levels for at least 2 years and these benefits were lost upon discontinuation. Thus, the current studies provide crucial mechanistic and clinical insight into the beneficial effects of verapamil in T1D.

[1] Comprehensive Diabetes Center, University of Alabama at Birmingham, Birmingham, AL 35294, USA. [2] Department of Medicine, Division of Endocrinology, Diabetes, and Metabolism, University of Alabama at Birmingham, Birmingham, AL 35294, USA. [3] Department of Microbiology, University of Alabama at Birmingham, Birmingham, AL 35294, USA. [4] School of Nursing, University of Alabama at Birmingham, Birmingham, AL 35294, USA. [5] Department of Biomedical Sciences, College of Veterinary Medicine, Cornell University, Ithaca, NY 14853, USA. [6] Biological Sciences Division, Pacific Northwest National Laboratory, Richland, WA 99352, USA. ✉email: shalev@uab.edu

Diabetes continues to grow as a chronic global health problem, affecting people of all ages. Since the discovery of insulin, a century ago, therapies have improved dramatically, but many critical needs and hurdles remain and prevent subjects with diabetes from living a truly normal life. In the case of type 1 diabetes (T1D), which involves autoimmune and inflammatory processes and destruction of insulin-producing pancreatic beta cells, exogenous insulin is still the only available therapy, and it is associated with the inherent risk of low blood glucose levels or hypoglycemic episodes that can be life-threatening. In addition, administration of insulin entails either continuous infusion via a pump or multiple daily injections and no oral medications are available. Recently, we reported the results of a small, phase 2, randomized placebo-controlled trial, using oral verapamil, an approved blood pressure medication, in new-onset T1D subjects[1]. Subjects receiving verapamil had improved endogenous beta cell function (as measured by a 2 h mixed-meal-stimulated C-peptide area under the curve (AUC)), lower insulin requirements, and fewer hypoglycemic events as compared to individuals getting placebo added to their standard insulin regimen[1]. Even though all subjects were normotensive, verapamil did not lead to any hypotension or any adverse events. While highly promising, these findings also raised a number of new mechanistic questions, including what exact biological changes verapamil elicits in humans with T1D, how long they may last, and how these changes and any potential associated therapeutic success could be monitored. The current studies were aimed at addressing these questions using proteomics, transcriptomics, and pathophysiological approaches. In this work, we now show that verapamil reverses T1D associated increases in serum CHGA levels, proinflammatory interleukin-21 (IL-21) levels, and T-follicular-helper (Tfh) cell markers and promotes an anti-oxidative, anti-apoptotic and immunomodulatory gene expression profile in human islets. In addition, our results suggest that continuous use of oral verapamil in subjects with T1D may delay loss of beta cell function and lower insulin requirements for at least 2 years post-diagnosis and that such therapeutic success or disease progression can be monitored by changes in serum CHGA.

## Results

To assess potential systemic changes in response to verapamil treatment, we conducted a global proteomics analysis using liquid chromatography-tandem mass spectrometry (LC-MS/MS) of serum samples from subjects at baseline and after 1 year of receiving verapamil or placebo and determined the effect of treatment on changes over time. 10 subjects had sufficient usable serum for LC-MS/MS at both of these time points resulting in 20 samples used for proteomics analysis (Supplementary Fig. 1). The baseline subject characteristics of this subset and of the full set of study participants are shown in Supplementary Table 1 and they demonstrate that there were no significant differences between the treatment groups with respect to age, gender, race, BMI, or HbA1C. Also, analysis of serum from the same subject before and 1 year after treatment allowed each subject to provide its own baseline control and avoid confounding effects from inter-individual variability. Applying a global serum proteomics workflow (Supplementary Fig. 1) along with this tightly controlled design, we were able to quantify 31,457 unique peptides and 867 proteins (<1% false discovery rate, FDR) with TMT reporter ion intensity data across all channels without missing data in the 20 serum samples (Supplementary Fig. 2). The raw dataset from this study is available at ProteomeXchange (accession # PXD026601). Following statistical analysis on treatment effect using a linear regression model, 53 proteins were identified

whose relative abundance over time was significantly altered ($P < 0.05$) by verapamil treatment (Supplementary Table 2). Enrichr[2] analysis of these proteins revealed enrichment for gene ontology biological processes such as neutrophil-mediated immunity and regulation of acute inflammatory and humoral immune responses as well as regulation of cellular metabolic processes (Supplementary Table 3). In fact, we observed a downregulation of leukocyte immunoglobulin-like receptor subfamily A member 3 (LILRA3 aka CD85e) and of secreted and transmembrane protein 1 (SECTM1) in response to verapamil, both proteins known to be involved in immune modulation (Supplementary Table 2). The cluster of differentiation 81 (CD81), which associates with CD4 and CD8 on T cells and provides a costimulatory signal with CD3 was also downregulated by verapamil, as was osteoclast-associated immunoglobulin-like receptor (OSCAR), a member of the leukocyte receptor complex protein family that regulates innate and adaptive immune responses. Interestingly though, chromogranin A (CHGA) emerged as the top serum protein altered by verapamil treatment in subjects with T1D exhibiting the most significant change in relative abundance over time in response to treatment as assessed by linear regression (Supplementary Table 2). In addition, two-sided t-test and paired t-test demonstrated that CHGA (unlike most other proteins) was also significantly downregulated in the verapamil group at year 1 as compared to placebo ($P = 0.007$) and as compared to baseline ($P = 0.004$), respectively. CHGA is a secreted glycoprotein produced by neuroendocrine cells and as such, it has been well established as a marker of neuroendocrine tumors. Within the cells, CHGA is localized in secretory granules including those of pancreatic beta cells, raising the possibility that changes in its circulating levels might also reflect alterations in beta cell integrity.

Indeed, while on average the relative abundance of CHGA in serum did not significantly change during the first year of T1D in control subjects receiving placebo (Fig. 1a), it decreased significantly in each subject receiving verapamil (Fig. 1b). Moreover, comparison of these baseline to year 1 changes in the verapamil group as compared to the control group also revealed a clear and significant difference between the study groups (Fig. 1c). To further confirm these LC-MS/MS results, we also measured serum levels of CHGA by ELISA in the full set of study subjects (Supplementary Table 1). The results were very much in alignment and again revealed no significant change over time in controls (Fig. 1d), but a significant and consistent decrease in serum CHGA after 1 year of verapamil treatment (Fig. 1e). Comparison of these baselines to year 1 changes in the verapamil group versus the control group again demonstrated a striking and significant difference between the study groups (Fig. 1f). Interestingly, serum CHGA levels in healthy, non-diabetic volunteers were ~2-fold lower as compared to those in subjects with T1D, but after 1 year of verapamil treatment, there was no longer any significant difference between verapamil-treated T1D subjects and healthy individuals (Fig. 1g). Next, we compared these results to available mixed-meal-stimulated C-peptide AUC data from the same subjects. The stimulated C-peptide AUC has remained the gold standard for assessing pancreatic beta cell function in T1D, with a decrease indicating disease progression, whereas stable levels or an increase are considered signs of successful therapeutic intervention[3–5]. In fact, serum CHGA showed a significant inverse correlation with C-peptide AUC (Fig. 1h). Moreover, C-peptide AUC decreased in each individual receiving placebo, while trending up in subjects getting verapamil (Supplementary Fig. 3a, b), providing a mirror image of the changes in CHGA observed in the same subjects (Fig. 1a, b). In addition, individual longitudinal changes in CHGA inversely correlated with individual longitudinal changes in C-peptide AUC (Supplementary

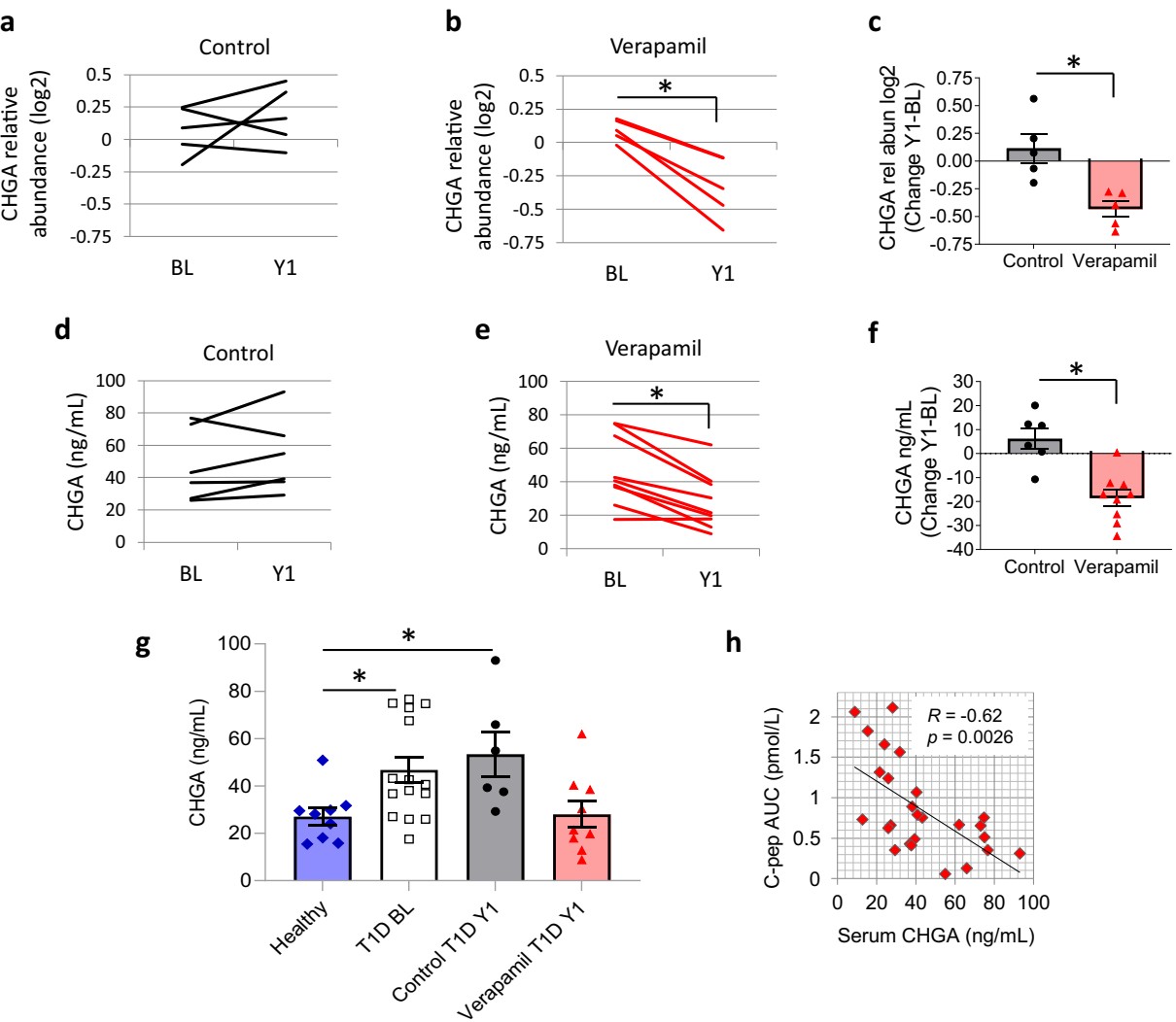

**Fig. 1 Serum CHGA in response to verapamil treatment of subjects with T1D.** CHGA as assessed by LC-MS/MS ($y$-axis represents relative abundance levels in zero-centered log2 form) in serum at baseline (BL) or at 1 year (Y1) of individual control subjects with T1D receiving placebo (black) ($n = 5$) (NS) (**a**) or verapamil (red) ($n = 5$) ($t_4 = 5.966$, *$P = 0.0040$) (**b**). Comparison of the changes in CHGA (BL to Y1) as assessed by LC-MS/MS in the verapamil and the placebo group ($t_8 = 3.674$, *$P = 0.0063$) (**c**). Serum CHGA levels at BL or Y1 as assessed by ELISA in individual control subjects with T1D receiving placebo ($n = 6$) (NS) (**d**) or verapamil ($n = 9$) ($t_8 = 5.44$, *$P = 0.0006$) (**e**). Comparison of the changes in CHGA (BL to Y1) as assessed by ELISA in the verapamil and the placebo group ($t_{13} = 4.497$, *$P = 0.0006$) (**f**). Serum CHGA levels in healthy, non-diabetic volunteers (blue) ($n = 9$) as assessed by ELISA and compared to subjects with T1D at baseline (T1D BL) (white) ($n = 15$), subjects with T1D getting placebo for 1 year (Control T1D Y1) ($n = 6$) or receiving verapamil for 1 year (Verapamil T1D Y1) ($n = 9$) ($F_{3,35} = 4.392$, *$P = 0.01$) (**g**). Correlation of C-pep AUC and serum CHGA ($R = -0.62$, $P = 0.0026$) (**h**). Bars represent means ± SEM. For **a**, **b**, **d**, **e**, two-tailed, paired Student's $t$-test. For **c**, **f**, two-tailed Student's $t$-test. For **g**, one way ANOVA and for **h**, repeated measures correlation coefficient by mixed model. Subject characteristics are listed in Supplementary Table 1. Source data are provided as a Source Data file.

Fig. 3c). Thus, serum CHGA seems to reflect changes in beta cell function in response to verapamil treatment or T1D progression and therefore may provide a longitudinal marker of treatment success or disease worsening. This notion is supported by recent reports suggesting that CHGA may serve as a biomarker for some autoimmune diseases including T1D[6,7]. Also, testing for CHGA only requires a simple blood draw and therefore may provide an easy and straightforward way to monitor changes in response to therapy or T1D progression over time. This would address a critical need, as the lack of a simple longitudinal marker has been a major challenge in the T1D field.

Therefore, to determine whether longitudinal changes in serum CHGA would continue to mirror therapeutic effects or disease progression over a longer period of time, we measured CHGA levels in a small number of study subjects with T1D who had received verapamil in year 1 and continued the treatment for a second year after diagnosis, as well as in control study participants who had never received verapamil. In addition, a subset of verapamil users discontinued the treatment after completion of the 1-year study, and their serum CHGA was analyzed as well. The subject characteristics of these subgroups are shown in Supplementary Table 4. Interestingly, we found that CHGA levels continued to decline and remained lower over the 2 years of verapamil treatment as compared to control subjects just receiving standard T1D treatment (Fig. 2a). Most strikingly, CHGA levels rose in those subjects that discontinued verapamil in year 2 (Fig. 2a). Inversely, C-peptide AUC remained stable over the 2-year period in subjects taking verapamil, whereas it continued to decline during the second year in the control group (Fig. 2b). Moreover, discontinuation of verapamil led to a sharp

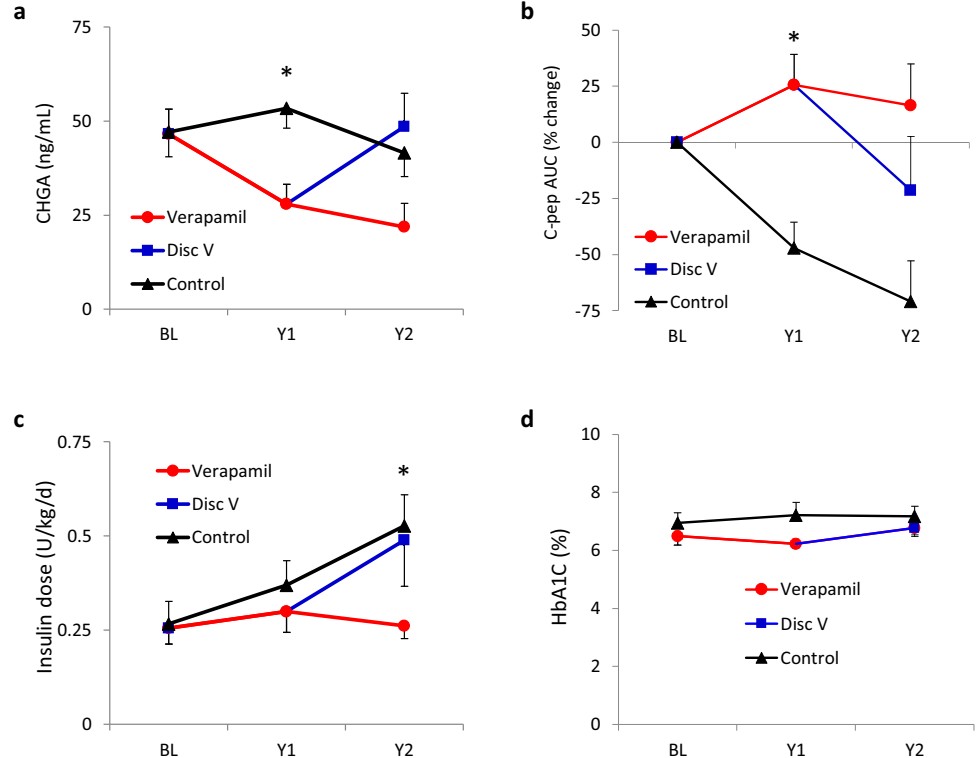

**Fig. 2 Insulin requirements, beta cell function, and CHGA over 2 years of T1D treatment with verapamil.** Changes over time in serum CHGA as assessed by ELISA ($F_{4,19} = 8.723$, $P = 0.0003$) (**a**), C-peptide AUC ($F_{4,19} = 4.346$, $P = 0.012$) (**b**), daily insulin dose ($F_{4,23} = 3.094$, $P = 0.036$) (**c**) and blood glucose control as assessed by HbA1C (**d**) in subjects with T1D receiving verapamil for 2 years (Verapamil; $n = 5$), discontinuing verapamil after the first year (Disc V; $n = 4$), or not taking any verapamil (Control; $n = 6$). Means ± SEM are shown, two-way repeated-measures ANOVA. Subject characteristics of these subgroups are shown in Supplementary Table 4. Source data are provided as a Source Data file.

drop in C-peptide AUC during year 2 (Fig. 2b). Consistent with these changes, the insulin dose required to control blood glucose levels remained low and stable over the 2-year period with verapamil treatment but continued to increase in the control group (Fig. 2c). Also, discontinuation of verapamil resulted in a clear increase in insulin requirements during year 2 (Fig. 2c). Blood glucose control as assessed by HbA1C remained stable over the 2-year period and was similar in the different study groups (Fig. 2d). The corresponding individual data are shown in Supplementary Fig. 4a–d, respectively. Together, these results not only show that changes in CHGA continue to reflect alterations in beta cell function over time providing an attractive longitudinal marker, they also demonstrate that with continuous use the beneficial effects of verapamil in subjects with T1D persist for at least 2 years and that verapamil treatment effectively keeps exogenous insulin requirements low.

Intriguingly, CHGA has also been identified as an autoantigen in T1D and one of its peptide fragments has been reported to be recognized as an epitope by diabetogenic T-cells[8,9]. Together with our global serum proteomics results and our discovery that verapamil effectively lowered serum CHGA levels and resulted in persistent beneficial effects in the context of autoimmune T1D, this raised the question of whether verapamil might also have any effects on T-cells. We, therefore, analyzed T1D-induced changes in T-cell markers using peripheral blood monocytes (PBMCs) from the same study participants with T1D whose serum was analyzed by LC-MS/MS and compared them to available PBMCs from non-diabetic healthy volunteers (Supplementary Table 1). General markers of CD4 T-helper (Th) cells and of proinflammatory Th1 cells such as C-X-C chemokine receptor type 3 (CXCR3, aka CD183) and signal transducer and activator of

transcription 4 (STAT4), were not significantly altered in subjects with T1D as compared to healthy controls and were not affected by verapamil treatment (Fig. 3a–c). In contrast, expression of CXCR5 (aka CD185), a surface marker of proinflammatory T follicular helper (Tfh) cells, and the Tfh signature cytokine, interleukin 21 (IL21) were significantly elevated in PBMCs of subjects with T1D as compared to healthy controls and these changes were reversed by verapamil treatment (Fig. 3d, e). Of note, serum IL-21 levels followed the same pattern, revealing again significantly elevated levels in T1D as compared to healthy controls and a ~2-fold reduction in response to verapamil treatment (Fig. 3f). Absolute serum IL-21 levels in healthy controls were comparable to those of healthy adults reported previously[10,11]. Also, the comparison of individual changes in these markers showed a similar trend with verapamil preventing the increase in CXCR5 from BL to Y1 and leading to a significant decrease in serum IL21 (Supplementary Fig. 5). This apparent increase in Tfh cell markers in subjects with T1D is consistent with recent reports of a Tfh cell signature in T1D that also included elevation in circulating Tfh cells and IL-21[12–15]. In addition, these changes have been suggested to play a role in the pathogenesis of T1D and to be potentially amenable to interventions[12–15]. Now our results reveal for the first time that verapamil treatment can reverse these T1D-induced changes. This suggests that verapamil, and/or the T1D improvements achieved by it, can modulate some proinflammatory cytokines and Th cell subsets, which in turn may contribute to the overall beneficial effects observed clinically.

The fact that these clinical improvements included an increase in C-peptide AUC and preservation of endogenous beta cell function suggested that verapamil might also directly affect

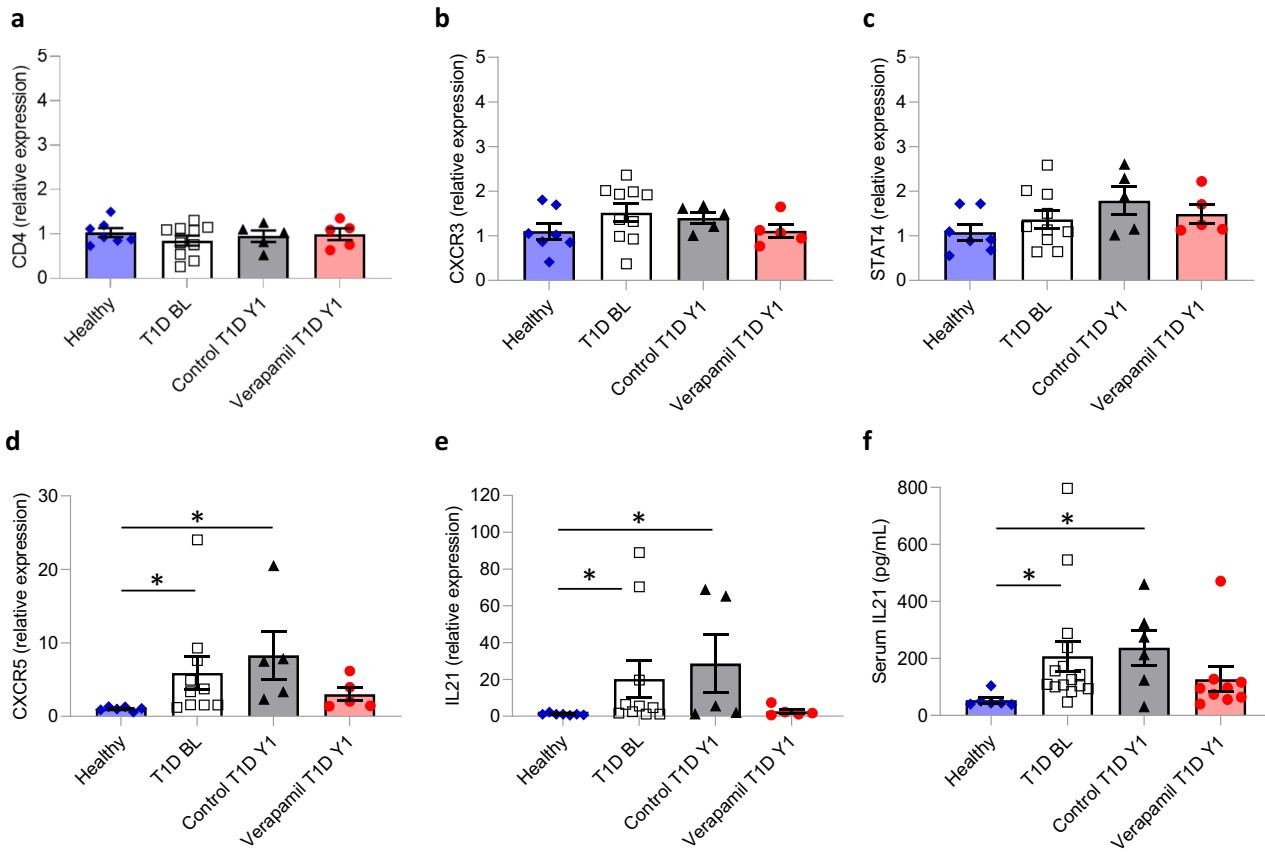

**Fig. 3 Effects of T1D and verapamil treatment on T-cells.** Expression of the T-cell markers CD4 (NS) (**a**), CXCR3 (NS) (**b**), STAT4 (NS) (**c**), CXCR5 ($H_3 = 14.011$, *$P = 0.003$) (**d**) and IL21 ($H_3 = 11.516$, *$P = 0.009$) (**e**) as assessed by qPCR in PBMCs from healthy, non-diabetic volunteers (blue) ($n = 7$), subjects with T1D at baseline (T1D BL) (white) ($n = 10$), subjects with T1D getting placebo for 1 year (Control T1D Y1) (grey) ($n = 5$) or receiving verapamil for 1 year (Verapamil T1D Y1) (red) ($n = 5$). Serum levels of the proinflammatory cytokine IL-21 as assessed by ELISA ($H_3 = 11.847$, *$P = 0.008$) (**f**). Bars represent means ± SEM; Kruskal-Wallis (nonparametric ANOVA) and Dunn's multiple comparisons. Source data are provided as a Source Data file.

pancreatic islets. We, therefore, next investigated the effects of verapamil on the overall gene expression profile by conducting RNA sequencing of verapamil-treated ($n = 3$) or untreated ($n = 3$) human islet samples from three different donors each serving as its own control. The dataset generated during the current study is available in the GEO repository (accession # GSE181328). Indeed, this transcriptomics analysis revealed that a large but comparable number of genes was up- as well as downregulated in response to verapamil (907 up and 619 down, respectively) as shown in the volcano plot (Fig. 4a). Further global analysis of these genes by gene ontology showed enrichment for a variety of biological processes, but interestingly three of the top ten processes were related to neutrophil-mediated immunity, degranulation, and activation (Supplementary Table 5). Of note, these are the same three processes identified in the Enrichr analysis of the human serum proteomics results (Supplementary Table 3), further supporting the notion that verapamil may also have some immune-modulatory functions. Recently, the notion of pancreas-resident or infiltrating neutrophils has been established[16] and some of the leucocyte cell surface molecules included in the neutrophil-associated processes identified by Enrichr have previously been found to be expressed in isolated pancreatic islets and islet endothelial cells[17,18]. This makes it tempting to speculate that changes in these islet-resident cells may have contributed to this surprising signature in the islets. Moreover, another process that was identified by multiple enriched terms, was antigen processing and presentation by major histocompatibility complex (MHC) class I molecules (Supplementary Table 5). Intriguingly, upregulation of islet MHC class I and class II

antigen expression has been suggested as a defining feature of T1D and as a marker for the associated interferon alpha response and inflammation[19–21]. In alignment with the normalization of proinflammatory markers found in response to verapamil, we observed downregulation of genes encoding human leucocyte antigen (HLA)-A, HLA-B, HLA-C, HLA-G (MHC class I) as well as HLA-DPA1 and HLA-DRA (MHC class II) (Fig. 4b–d). In terms of individual genes, the most extremely upregulated gene in response to verapamil was insulin-induced gene 1 (INSIG1) (Fig. 4a), which has previously been reported to promote anti-apoptotic BCL2 and reduce beta cell apoptosis[22], consistent with the observed protective effects of verapamil. The most significantly downregulated gene was pancreatic secretory granule membrane major glycoprotein 2 (GP2) (Fig. 4a). GP2 is a specific cell surface marker of human pancreatic progenitors, has been associated with increased risk of type 2 and gestational diabetes, and has generally been identified as an immunomodulator[23–25]. Genes with the next most significant changes in expression in response to verapamil included the upregulated enzymes methylsterol monooxygenase 1 (MSMO1), isopentenyl-diphosphate delta isomerase 1 (IDI1), and squalene epoxidase (SQLE) as well as downregulated lysyl oxidase homolog 4 (LOXL4), actin alpha 2 (ACTA2), and glycerol-3-phosphate dehydrogenase (GPD1). In addition, several genes that have previously been shown to modulate key islet processes including oxidative stress, apoptosis, and T1D autoimmunity were significantly up- or downregulated by verapamil (Fig. 4a, b). Of note, the expression of thioredoxin-interacting protein (TXNIP), which we have previously found to be downregulated by verapamil

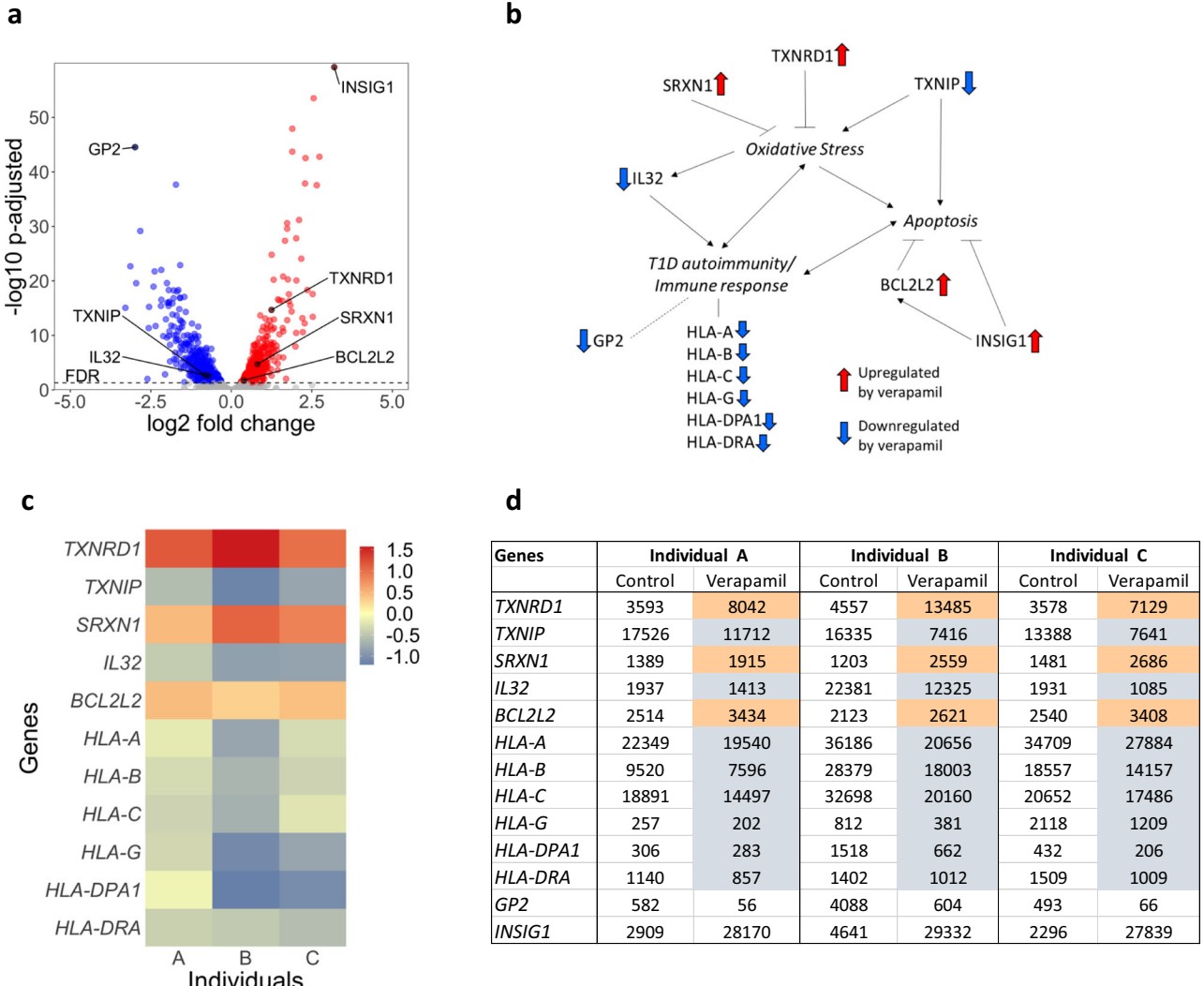

**Fig. 4 Gene expression profile changes in human islets in response to verapamil treatment.** RNA sequencing was performed on isolated human islets from three different individuals (A–C) treated for 24 h with or without verapamil (100 μM) with each donor serving as its own control. Volcano plot contains all genes with a baseMean expression of >500. Those genes with an adjusted DESeq2 *P*-value < 0.05 (calculated using a Wald test and the Benjamini–Hochberg method) are shown in blue (downregulated) and red (upregulated) (**a**). Key pathways modulated by differentially expressed genes (**b**). Heatmap showing key genes changed after treatment with verapamil (color scale represents log2 fold change) (**c**). Display of the normalized read counts for key genes before and after verapamil treatment of islets from each of the individual islet donors A–C (**d**).

in vitro and in vivo[26] was consistently decreased in response to verapamil in all samples as shown in the heatmap (Fig. 4c) and by the normalized read counts (Fig. 4d). In fact, TXNIP is considered a key factor in diabetes-associated beta cell apoptosis[27–29] and genetic deletion of TXNIP has been shown to mimic the anti-diabetic effects of verapamil in different mouse models[26,29]. Downregulation of TXNIP has therefore been suggested to mediate the beneficial effects of verapamil in the context of diabetes[26] and this notion is strongly supported by the current findings in human islets. TXNIP belongs to the thioredoxin network, a cellular redox system, however, by interacting with and inhibiting thioredoxin, TXNIP promotes oxidative stress. In contrast, thioredoxin reductase (TXNRD1) and sulfiredoxin (SRXN1), two additional members of the thioredoxin signaling network, reduce oxidized thioredoxin and peroxiredoxin back to their active states and thereby preserve the cellular redox potential. In alignment with the overall protective effects, the expression of *TXNRD1* and *SRXN1* was significantly upregulated by verapamil (Fig. 4a–d). In addition, the gene encoding Bcl-2-like protein 2 (*BCL2L2*), a pro-survival member of the bcl-2 protein family was also significantly

upregulated in response to verapamil (Fig. 4a–d). This is consistent with the anti-apoptotic effects found with TXNIP downregulation and with verapamil treatment in diabetic mouse models[26,29]. Furthermore, verapamil downregulated the expression of interleukin 32 (*IL32*) (Fig. 4a–d), a unique proinflammatory cytokine found only in primates that is induced by oxidative stress[30] and has recently been suggested to be upregulated in pancreatic islets and play a role in T1D autoimmunity[31,32].

## Discussion

In summary, the results of these exploratory studies suggest that continuous use of oral verapamil in individuals with T1D may delay disease progression and lower insulin requirements for at least 2 years post-diagnosis and that this is associated with normalization of serum CHGA levels as well as of proinflammatory IL-21 levels and Tfh cell markers. In addition, they show that verapamil regulates the thioredoxin system and promotes an anti-oxidative, anti-apoptotic and immunomodulatory gene expression profile in human islets suggesting that together these

protective changes might explain the overall beneficial effects observed with verapamil.

The significantly lower insulin requirements found after 2 years of verapamil treatment as compared to controls is consistent with earlier findings after 1 year of treatment[1] and with the observed preservation of beta cell function. In addition, our results showing verapamil regulating the thioredoxin system and inhibiting TXNIP expression in the islets provide a potential mechanistic explanation for these beta cell sparing effects. This especially when considering the beta cell protective and anti-diabetic effects observed in different mouse models in response to genetic TXNIP deletion or verapamil-induced TXNIP inhibition[26,29]. Moreover, in humans with T1D even a small amount of preserved endogenous insulin production as opposed to higher exogenous insulin requirements has been shown to be associated with improved outcome[33] and could help improve quality of life and lower the high costs associated with insulin use. The fact that these beneficial verapamil effects seemed to persist for 2 years, whereas discontinuation of verapamil led to disease progression, provides some additional support for this approach and its potential usefulness for long-term treatment. Intriguingly, global proteomic profiling of this unique set of before and after treatment samples also led to the discovery of serum CHGA as a potential therapeutic marker. Serum CHGA is a simple and easy blood test, showed good correlation with loss of beta cell function, accurately reflected changes in response to verapamil therapy or treatment discontinuation and as a major advantage, persisted over a time period of at least 2 years. Furthermore, due to its role as a T1D antigen[8,9], it is tempting to speculate that the lowering of CHGA in response to verapamil might even help dampen some of the T1D autoimmunity. In any case, our results reveal for the first time that verapamil may reverse specific T1D-induced T-cell changes and thereby may also modulate certain aspects of the immune response. Intriguingly, Tfh cells and IL-21, both suggested to be downregulated by verapamil, have recently been reported to play an important role in the autoimmunity of T1D[12–14]. In fact, this may help explain why verapamil treatment was so successful even in the absence of any additional bona fide immunomodulatory intervention. This provocative idea of additional immune modulatory effects is supported by the observed enrichment in gene ontology processes associated with MHC class I antigen presentation and neutrophil-mediated functions in response to verapamil. Thus, the present exploratory studies provide the first indication of some potentially sustained benefits of verapamil use in the context of T1D. They further suggest that verapamil may result in protective effects not only at the level of insulin-producing islet beta cells, but also at the level of T-cells and proinflammatory cytokines, uncovering a previously unappreciated connection between verapamil use and the immune system in T1D. However, since the current studies were based on a very small subset of subjects, these initial findings will have to be confirmed in larger studies such as the ongoing Ver-A-T1D (NCT04545151) or CLVer (NCT04233034) verapamil trials. In addition, long-term trials with extensive sample analysis using this new knowledge will ultimately have to validate CHGA as a proper biomarker as well as the novel clinical and mechanistic insights gained in the current studies.

## Methods

**Human subjects**. All human studies were approved by the University of Alabama at Birmingham (UAB) Internal Review Board and written informed consent was obtained from all participants in accordance with the criteria set by the Declaration of Helsinki. Participants' compensation was $80 per completed MMTT. Subjects with T1D had been diagnosed within 3 months and were positive for T1D associated auto-antibodies. All continued on their standard insulin regimen but were on no other diabetes medications during the entire 2 years. They were taking randomly assigned verapamil (360 mg sustained-release daily) or placebo in a

blinded fashion for one year as described in the protocol and consort table of the initial trial [clinicaltrials.gov/ct2/show/NCT02372253 2/20/2015][1] and then chose to be on or off verapamil and to be followed for a second year. This resulted in five subjects who continued to receive verapamil (360 mg sustained-release daily) for 2 years, four subjects who discontinued active study drug for the second year, and six participants randomized to the control arm who never received active study drug over the 2-year period. All other subjects declined the second-year follow-up and/or had no usable blood samples. The study subject demographic characteristics at baseline and at the start of year 2 are listed in Supplementary Tables 1 and 4, respectively. Supplementary Table 6 provides an additional overview of all the individual study participants, use of their samples, and their clinical data. Healthy volunteers were non-diabetic as confirmed by HbA1c, had not been diagnosed with any illness, were not receiving any prescription medications, and never received any study drug. Their characteristics are also shown in Supplementary Table 1 even though they only provided blood samples used as a comparison for some of the measurements of this exploratory study.

Remaining beta cell function was assessed using stimulated C-peptide AUC during a mixed-meal tolerance test (MMTT) as described previously[4,5]. The MMTT was only performed when fasting blood glucose levels were within the range of 3.9–11.1 mmol/L and otherwise, the test was rescheduled. Blood samples were collected at −10, 0, 15, 30, 60, 90, and 120 min for serum C-peptide. The mean C-peptide AUC (0–120 min) was calculated using the trapezoidal rule[5] and the percent change from baseline was determined for each individual. In addition, the daily insulin dose required was calculated by analyzing the patient's mean daily insulin use during a 2-week period at baseline, year 1 and year 2. Glycemic control was monitored by measurements of HbA1c. Serum samples and PBMCs were collected and stored at −80 °C until further analysis.

**Proteomics/liquid chromatography-tandem mass spectrometry (LC-MS/MS)**. 20 serum samples from 10 subjects collected for each subject at baseline and after 1 year of receiving verapamil or placebo were analyzed using a standardized workflow (Supplementary Fig. 1) similarly as previously reported[34]. For sample processing, 40 μL of serum was subjected to immunodepletion of the most abundant serum proteins using a MARS Hu-14 column (Agilent Technologies, Palo Alto, CA) as previously described[35]. The flow-through fractions were concentrated using Amicon spin filters with 3 kDa molecular mass cutoffs (Millipore, Burlington, MA). Protein concentration was measured by BCA assay (Thermo Scientific, San Jose, CA) prior to protein digestion. The depleted samples were then digested by a urea-based protocol and the peptides were then desalted by solid-phase extraction (SPE) (Phenomenex, Torrance, CA) and dried in a vacuum centrifuge. 100 μg of peptides of each sample were labeled using 11-plex tandem mass tag (TMT) reagents (Thermo Fisher Scientific, Waltham, MA) following the recent published protocol[36]. One pooled sample was generated by pooling an aliquot of 25 μg peptides from each sample to serve as a "universal reference". The reference sample was included as the 11th channel for the two TMT-11 experiments (Supplementary Fig. 1). The TMT-labeled peptides combined from all 11 channels were further fractionated by basic pH reversed-phase LC using a C18 column (250 mm × 2.1 mm, 5 μm particles, Waters, Milford, MA) using an Agilent 1200 HPLC. Ninety-six fractions were collected and concatenated into 24 fractions as previously described[37], dried in a vacuum centrifuge, and resuspended in 0.1% formic acid.

For LC-MS/MS analysis, fractionated peptide samples were analyzed using a nanoAquity UPLC system (Waters) coupled to an Orbitrap Fusion Lumos mass spectrometer (Thermo Fisher Scientific). LC separations were performed with a custom-packed analytical C18 column (50 cm × 75 μm i.d., 3 μm particle size of Jupiter C18, Phenomenex) with a 120 min gradient. Binary mobile phases comprised of buffer A (0.1% formic acid in water) and buffer B (0.1% formic acid in acetonitrile) were used at a flow rate of 300 nl/min. For peptide elution, the percentage of buffer B was increased linearly and a 10 min wash with 95% buffer B and a final 1 min wash with 100% buffer B was also included. Obitrap full MS scans were conducted from 350 to 1800 $m/z$ with a resolution of 60 K and AGC target of $4 \times 10^5$ followed by data-dependent higher-energy collisional dissociation (HCD) MS/MS acquisitions at a resolution of 50 K (AGC $1 \times 10^5$) and a maximum injection time (IT) of 105 ms for a total cycle time of 2 s. The MS/MS isolation window was set as 0.7 m/z with HCD normalized collision energy of 32. Peptide mode was selected for monoisotopic precursor scan and charge state screening was enabled to reject unassigned 1+, 7+, 8+, and >8+ ions with a dynamic exclusion time of 45 s to discriminate against previously analyzed ions between ± 10 ppm.

For data processing and analysis, the thermo RAW files were first processed with mzRefinery to characterize and correct for any instrument calibration errors, and then with MS-GF+ v2109.08.26[38] to match against the UniProt human proteome database (2019.11.05 release; 20,352 entries). A decoy database from the searched fasta files was created by MS-GF+ to estimate the FDR. As searching parameters, the parent ion mass tolerance was set at 20 ppm, 2 missed cleavages were allowed. Cysteine carbamidomethylation (+57.0215) and N-terminal/lysine TMT labeling (229.1629) were searched as static modifications, whereas methionine oxidation (15.9949) was set as variable modification. Spectral-peptide matches were filtered using PepQValue < 0.005 and <7 ppm resulting in maximum FDR of 1%. A minimum of 6 unique peptides per 1000 amino acids of protein length was then required for achieving 1% at the protein level within the full data set. Post-processing of quantitative data of TMT reporter ion intensities was

performed using a R package "PlexedPiper" for isobaric quantification [https://github.com/PNNL-Comp-Mass-Spec/PlexedPiper] similar as previously reported[39]. Briefly, the intensities of all 11 TMT reporter ions were extracted using MASIC software (v 3.0.7235) [https://github.com/PNNL-Comp-Mass-Spec/MASIC/][40]. The reporter ion intensities from different scans and different fractions corresponding to the same gene were grouped. Relative protein abundance was calculated as the ratio of abundances between individual sample channels to the reference channel using the summed reporter ion intensities from peptides that could be uniquely mapped to a gene. The relative abundances were log2 transformed and zero-centered for each protein to obtain final relative abundance values. Statistical analyses to test protein abundance changes over time due to either drug effect or progression of T1D were performed using a mixed effects model, where treatment group and timepoint factors were modeled as fixed effect and subjects were modeled as random effect. The significance of the drug treatment was tested using interaction between the group and timepoint effects. Thus, the full model was formulated as $protein \sim timepoint{:}group + timepoint + group + (1|subject)$. The significance of the $timepoint{:}group$ was tested using the nested model approach. The test was two-tailed and was performed using $lme4$ package [doi:10.18637/jss.v067.i01] of R language for statistical computing [https://www.R-project.org/]. Proteins with significant changes in abundance were further analyzed using Enrichr Gene Ontology Biological Process 2021 term enrichment[2].

**ELISA**. Serum CHGA levels were assessed using the Human Chromogranin A ELISA Kit (Epitope Diagnostics, INC., San Diego, CA). Serum IL-21 levels were measured using the Human IL-21 Proquantum Immunoassay Kit (Thermo Fisher Scientific) according to the manufacturer's instructions (limits of quantitation: 0.32–5000 pg/mL).

**Quantitative real-time PCR**. Total RNA was extracted from PBMCs using the miRNeasy Mini Kit (Qiagen) according to the manufacturer's instructions. RNA was reverse transcribed to cDNA using the first strand cDNA synthesis kit (Roche) and quantitative real-time PCR (qPCR) was performed on a LightCycler 480 system (Roche) as reported previously[41]. Relative expression of the proinflammatory T-cell markers CD4, CXCR3, STAT4, CXCR5, and IL21 was measured using the primers listed in Supplementary Table 7. Assessment of markers such as IL21 by qPCR using PBMCs has been successful in the past[12,42,43]. All results were normalized for GAPDH run as an internal standard and serial template dilutions were performed to confirm comparable target and reference amplification efficiency. The data was then analyzed using the 2^(-ddCT) method as previously described in detail[44].

**Transcriptomics/RNA sequencing**. Human islets from three different donors were obtained from the Integrated Islet Distribution Program (IIDP) and after overnight incubation at 5 mM glucose, 250 islets were handpicked per sample and incubated for 24 h in 25 mM glucose RPMI 1640 medium with or without 100 μM of verapamil prior to RNA extraction using a miRNeasy Mini Kit (Qiagen, Germantown, MD). This concentration of verapamil has been established in early studies for the use in in vitro experiments using beta cells and islets[45,46].

RNA sequencing was performed by Exiqon/Qiagen and included preparation of libraries using TruSeq stranded mRNA sample preparation kit (Illumina Inc., San Diego, CA) and single-end sequencing was performed with an average of ~43 million reads obtained per sample.

For data processing and analysis, RNA-sequencing reads were aligned to the H. sapiens reference genome (GRCh38.p7) using STAR (v2.4.2a) with an average ~90% reads uniquely mapped. Alignments were quantified using Salmon (v0.8.2) and differential expression analysis was performed using DESeq2. The DESeq2 model accounted for the experimental design of paired treated and untreated samples from each individual. DESeq2 was used to determine the significance of the differential expression. FDR was calculated using the Benjamini–Hochberg method. Significantly downregulated genes were further analyzed using Enrichr Gene Ontology Biological Process 2021 term enrichment[2].

**Statistical analysis**. All available data were included in the analysis and no data from adequate samples were excluded. Statistician and experimenters were blinded to the study group allocation of samples. Population characteristics of study subjects were summarized as mean and standard errors (SEM) for continuous variables and frequencies for categorical variables. The group comparison of baseline measures was conducted using Chi-square test, Fisher's exact test, or t-test where appropriate. The normal distribution assumption was checked using Q–Q plots and nonparametric analyses were performed where appropriate. To evaluate the effects of time and group, two-way repeated-measures ANOVA was used. One-way ANOVA or Kruskal–Wallis followed by Dunn's multiple comparison testing was used to assess the significance between multiple groups. All tests were two-sided. Statistical analyses were performed using SigmaStat 4.0 and SAS 9.4 (Cary, NC).

**Reporting summary**. Further information on research design is available in the Nature Research Reporting Summary linked to this article.

## Data availability
The proteomics data that support the findings of this study have been deposited in ProteomeXchange with accession: PXD026601.

The MS raw datasets can also be found in the online repositories: Massive.ucsd.edu with accession: MSV000087598.

The transcriptomics data have been deposited in GEO with accession: GSE181328. Publicly available data sets used can be accessed at: [https://www.ncbi.nlm.nih.gov/assembly/GCF_000001405.33/] (H. sapiens reference genome GRCh38.p7) and [https://www.uniprot.org/proteomes/UP000005640] (UniProt human proteome database (2019.11.05 release).

Remaining data are available within the Article, Supplementary Information, or the Source Data provided with this paper. Source data are provided with this paper.

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

## Acknowledgements

This work was supported by grants from the National Institutes of Health to A.S. (R01DK078752 and Human Islet Research Network U01DK120379) and to W.J.Q. (DP3DK110844 and R01DK122160) and ADA Pathway Award to P.S. (1-16-ACE-47). The UAB Human Physiology Core was supported by NIH award P30DK079626. Human pancreatic islets were provided by the NIDDK-funded Integrated Islet Distribution Program (IIDP) (RRID:SCR_014387) at City of Hope, NIH grant UC4DK098085. Mass spectrometry proteomics experiments were performed in the Environmental Molecular Sciences Laboratory, Pacific Northwest National Laboratory, a national scientific user facility sponsored by the DOE under Contract DE-AC05-76RL0 1830.

## Author contributions

G.X. conducted the molecular analyses and generated the figures, G.X., L.A.T., and T.B.G. helped with sample processing and, F.O. and T.G. were responsible for the MMTTs and sample and data collection. J.C. helped with the human islet processing, H.M.T. provided immunological expertise and P.L. helped with the statistical analysis. M.K. and P.S. were responsible for the RNA sequencing and A.A.S., A.C.S., T.T.L., V.A.P., and W.J.Q. conducted the mass spectrometry studies and analyses. A.S. designed the studies, analyzed the results, and wrote the manuscript. All authors reviewed and approved the manuscript.

## Competing interests

The authors declare no competing interests.
