## [Peer Review File · Nature Communications]

Exploratory Study Reveals Far Reaching Systemic and Cellular Effects of Verapamil Treatment in Subjects with Type 1 DiabetesREVIEWER COMMENTS

Reviewer #1 (Remarks to the Author):

this is a very interesting paper exploring the effects/benefits of verapamil in new onset T1D. Previously 1 yr data has been presented and now a some of the patients were followed for an additional year, where some continued on verapamil unblinded and others stopped, and some remained controls

study suggest that a beneficial effect in maintained with reduction in the marker ChromograninA, improvement in c-peptide AUC during stimulation, lower insulin need and effects on immune cells and interesting those stopping verapamil had "relapse" in the measures

Comments

it should be described how these participants were selected from the original study with 24 patients, why was not all patients followed and analysed? lost to follow up or selection?

the original study was small, this is a subset, so very small which has to be recognized clearly, thus the risks of false positive findings are high

for the comparisons in fig 1 you compare levels at baseline and at 1 year, please provide a comparison of the change in one group vs change in the other group (BL to Y1 in placebo, compared to BL to Y1 in V group) or use baseline adjusted values when comparing end of study levels applies to all comparisons

in fig 1M a correlation between chromogranin and AUC is provided, but it is not clear which patients are included, is it both baseline and end of study visits or only all at baseline or? (please clarify and only include each subject once or you need to adjust for dependency

given the very small numbers it would be relevant if possible to discuss if a larger clinical trial assessing this is ongoing/planned

Reviewer #2 (Remarks to the Author):

Xu et al report on the impact of verapamil on type 1 diabetes through evaluation of clinical follow-up from previously reported clinical trial, samples from the trial, and in vitro studies of verapamil on islets. Key findings reported by the investigators were that verapamil decreased Tfh cells and IL21 and increased chromogranin A and that continuous use of verapamil stabilized beta cell function. Further, the investigators report that verapamil induced anti-oxidative and anti-apoptotic genes expression in human islets. The investigators conclude that the studies further support the benefits of verapamil and provide mechanistic insights.

Strengths: Novel observation re: changes in chromogranin A

Weaknesses:

Major: There are insufficient details regarding experimental and analysis methods and human subjects in this paper and methodologic issues that limit the ability to determine whether the author's conclusions are valid.

For the clinical results, there is no information about the study protocol for which data is presented about those in the active treatment arm who continued or did not continue to receive ongoing active study drug and those who were in the control arm at the conclusion of the primary trial. For the mechanistic results, there is no data as to whether the samples were masked before testing and analysis. Further, the text states that serum was available on 20 individuals, yet there are variable numbers of subjects with data for each of the figures. There is insufficient information about "healthy" and other T1D individuals used in some experiments/figures. There is no information about whether these are the same or different subjects for the different experiments. For the qPCR data, there is insufficient description of methods. The reference cited regarding qPCR methods refers back to another paper for methods in animal models from 2008. Data is expressed as relative expression

values which is common, but there is no information about absolute levels of each transcript, and no information about how the relative expression assessments are made. It is challenging to understand why the investigators report global STAT4 expression rather than investigating phosphorylated stat. Similarly, QPCR in PBMC is an unusual way to measure IL21. Information about the limit of quantitation for serum IL21 should be included in the methods. It is unclear why flow or cytoF was not used to evaluate CXCR3 or CXCR5 cells. It is unclear if the same three islet preparations were used for Figure 4 C and 4 D.

Other:

Figure 1A/B data is the same as shown in a different way without added value in Figure C/D. Thus, the latter are not needed. The same is true for Figure E/F data which is duplicated in Figures 1G/H.

Figure 1J shows a well-known feature of T1D, comparing C-peptide in healthy individuals and T1D (and no information as to who these samples are from), and does not add value to the manuscript.

Figure 1K/L represents data from a subset of individuals in the original clinical trial; thus, it is not clear what this figure is aiming to demonstrate.

Figure M has >25 data points. This suggests that these are multiple data points from the same individuals and thus other statistical approaches are needed.

Figure 2A has a limited number of individuals; data would be better shown with individual results. Figure 2C cannot be interpreted without more information about the protocol and the blinding and selection and distributions of subjects for the f/u trial. Figure 2D is redundant to 2C (which does not have information about the DISCV group). A box plot is inappropriate way to illustrate the HbA1c values in the two arms and it is also missing the DISCV group.

Figure 3: More information needed about precise methods used for RNA seq and qPCR studies and analysis. Why were only 5 people per group included in the qPCR analysis? Figure 3 E; the differences between control and verapamil treatment are dictated by 2 extreme values. Is there a relationship between serum IL21 levels and the relative RNA expression?

Figure 4: Information about methods for this experiment are lacking detail (dose and time of exposure to verapamil), impact of verapamil on non-islet tissue? How does the dose used in these in vitro experiments relate to the therapeutic dose in the clinical trials? If the same islets and experiment were used to generate the data, there is no added value to Figure 4C and 4D (i.e. measuring RNA by 2 approaches that would be expected to correlate)

Text:

It seems odd and out of place to characterize MMTT as “cumbersome for patients and health care providers”.

There are many well-established parameters for validation of biomarkers depending upon the intended use. While the CHGA data is intriguing, the author’s statements about the utility of the currently reported measure as compared to selected other measures is not well supported.

References 11-13 describe IL-21 production by CD4 T cells and/or association with Tfh cells. There is no reference to previous reports on serum IL-21 levels in T1D.

For the reader’s benefit, it would seem worthwhile to reference ongoing RCTs using verapamil in T1D.

Reviewer #3 (Remarks to the Author):

The authors previously published a study about the beneficial effects of Verapamil on new onset type 1 diabetes patients. Interestingly, Verapamil is an approved and generic drug currently used for high blood pressure. In this paper, the authors perform plasma proteomics on a small cohort at base line and after one year. They also perform gene expression studies on islets with Verapamil as well.

In the plasma proteomics study, they find the most significantly changed protein to be chromogranin A (CHGA), which is known to be produced in beta cells and which has recently been implicated as an auto-antigen in T1D. The authors discuss the potential roles of CHGA in the context of T1D. In the gene expression study, they focus on down regulation of the thioredoxin system and how this might be beneficial to beta cells.

Overall, I find this study quite interesting and the findings potentially significant. My major reservations are the size of the cohort, which consists of only 9 T1D subjects given the drug, 6 control T1D and 6 healthy controls. The plasma proteomics pipeline was not easy to follow for this reviewer, mainly due to the use of custom scripts, but the large issue here is that the plasma proteomics results are nowhere documented at all, except for log2 changes of the top significant proteins. As the authors used a quite involved set up with TMT labeling, depletion, a reference channel etc. there is ample opportunity for variation. This is hard to judge given that no data is available. However, from this reviewer's experience, it would be hit or miss with such a small number of cases. CHGA goes down for 4 of the 9 subjects in Fig. 1b (where are the others?), but the controls go up or down as well, both in Fig. 1A and Fig. 1e. In conclusion, the plasma proteomics study is potentially interesting but not convincing as it is.

The gene expression study is relatively standard as far as I can tell and the observed effects quite broad.

Altogether, the study lacks rigor that would allow it to carry the far reaching interpretations that the authors make.

In more detail:

p.1, title, line 1

- 'omics reveal ...'

- Omics analysis is not sufficient to fit the title ® see later explanations

p.2, Abstract, line 3-4

- 'exact biological changes verapamil elicits in humans with T1D, how long they may last'

- Addressing only biological changes on specific picked proteins
 - o Proteomics: one single protein in serum proteome (CHGA)
 - o RNAseq: 5 proteins picked
 - o Others: c-pep, T-cell markers (CD4, CXCR3, STAT4, CXCR5, IL21)
- Time scale of omic analysis: only 1 time point (1 year)
 - o Where is the time point of 3 month if you compare it to your paper before (Ovalle et al.)
 - o Why no serum proteomics on second year samples?
- Additionally, RNA seq with only emphasize on own parameters without telling more about the outliers

p.2, Abstract, line 3-4

- 'unbiased proteomics analysis'

- Unbiased proteomic analysis completely missing
- Seems like a targeted proteomics analysis in the main text description

p.2, Abstract, line 6

- 'effects of continuous verapamil use over a 2-year period'

- Only shown c-pep AUC and CHGA level (measured by ELISA?) for second year point

p.2, Abstract, line 10-11

- 'Our present results reveal that verapamil regulates the thioredoxin system and promotes anti-oxidative and anti-apoptotic gene expression profile in human islets, ...'

- Those results were already known and can't be claimed as novel in this abstract (Ovalle et al.; Xu et al.; Chen et al.)

p.3, results, line 1-2

- 'treatment groups were well balanced in terms of subject characteristics'
- Healthy group only has female participants and a BMI above 27 (vs. 24)
- Describe this gender bias in main text
- If so compare only females in T1D vs. females of healthy

p.3/4, results, line 21 & 1-2

- 'unbiased proteomics analysis' & '53 proteins or peptides whose relative abundance over time was significantly altered ...'
- Unbiased proteomic analysis completely missing
- ® need to provide introduction figure about proteomic data set like study design, measured protein numbers, CV values, ...
- Experimental description of proteomic dataset totally missing in main text and method description not sufficient
- No comment to any of the other significant proteins/peptides
- ® description about picking CHGA insufficient and looks like a targeted approach not an unbiased analysis
- If you want to address the question of exact biological changes you need to use an unbiased analysis and comment on other significant proteins
- How were people for serum proteomic analysis picked (only 5 out of 6/9) ® no description and explanation provided
- Serum proteomics analysis of healthy people totally missing ® gender bias?

p.5, results, line 1-2

- Fig 1 J-L: is verapamil BL vs. Y1 significant? Is healthy vs. verapamil BL/Y1 significant?

p.5, results, line 4

- 'a highly significant inverse correlation'
- too strong wording at a correlation of -0.603

p.6, results, line 21

- 'interleukin 21 (IL21) were elevated >10-fold or 100-fold ...'
- differences of IL21 only because of 2 outlier patients
- ® needs to be discussed and may need additional replicates to claim this topic

p.7, results, line 14

- 'this genomic analysis'
- RNA seq is transcriptomic analysis
- Global transcriptomic analysis again missing and no description of data set: significant transcripts, outliers, ...
- Introduction of RNAseq data set needed ® no description in main text
- Fig 4b: enrichment analysis needs to be displayed globally to address "exact biological changes"
- o This doesn't present an omic analysis
- Best would have been if matched experiment that omic analysis could have been compared to serum proteomics ® how is the correlation? ...

p.7, results, line 15

- 'was up- as well as downregulated'
- Specify numbers and discuss outliers a little
- cherry picking on those which were already know (low p-values, low log2 fold change)
- Does not reflect the aim of discussing exact biological changes and of an omic analysis

p.10/11, methods, Proteomics /

- 'desalted by solid-phase extraction (SPE)'

- What kind of SPE?

- 'one pooled sample was generated by pooling an aliquot of peptides from each sample to serve as a "universal reference"

- Description not sufficient: how was sample pooled, equally? Amount of peptides?

- Description of referencing missing

- 'the TMT-labeled peptides combined from all 11 channels ...'

- Which samples were combined in one TMT experiment?

- Description not enough

- 'data processing and analysis, ... mzRefinery ... MS-GF+ ...'

- MS raw data needs to be provided on PRIDE

- Data processing description not sufficient

- Using of a not standard software @ test data processing in gold standard software, e.g. MaxQuant

- Provide all parameters set in software, 1% FDR and PepQValue < 0.005 not sufficient: e.g. ppm limits, peptide length parameters, ions, ...

- Describe exactly filtering and post processing of dataset

p.11, methods, Genomics / RNA Sequencing

- 'Genomics / RNA sequencing'

- Dataset needs to be provided for reviewers

- RNA sequencing provides no information about genomics

- Data set description not sufficient

- FDR model not mentioned and described

p.15/16, Figure legends

- description of figure legends not sufficient to understand the figures @ need to describe everything in more detail

- labels of x axis and y axis not sufficient @ especially a-d: CHGA (log2) / CHGA: are those log2 values on raw values? Relative values? Needs a more precise description

- 2a How measured are CHGA values measured? by ELISA?

- 4: no description of p-value adjusted

POINT-BY-POINT RESPONSE TO REVIEWERS' COMMENTS

Reviewer #1 (Remarks to the Author):

this is a very interesting paper exploring the effects/benefits of verapamil in new onset T1D. Previously 1 yr data has been presented and now a some of the patients were folowed for an additional year, where some continued on verapamil unblinded and others stopped, and some remained controls study suggest that a beneficial effect in maintained with reduction in the marker ChromograninA, improvement in c-peptide AUC during stimulation, lower insulin need and effects on immune cells and interesting those stopping verapamil had "relapse" in the measures

We are grateful to Reviewer #1 for the overall very positive comments and helpful suggestions that have allowed us to clarify some remaining questions and further improve our paper.

Comments

it should be described how these participants were selected from the original study with 24 patients, why was not all patients followed and analysed? lost to follow up or selection?

Inclusion of participants in the current exploratory study was not based on any selection, but rather on patient consent to be followed for a second year and availability of usable samples for the tests to be conducted. We have now clarified this important point in the revised text.

the original study was small, this is a subset, so very small which has to be reckognized clearly, thus the risks of false positive findingsa are high

We have also further clarified this point in the discussion of our findings.

for the comparisons fex in fig 1 you compare levels at baseline and at 1 year, please provide a comparison of the change in one group vs change in the other group (BL to Y1 in placebo, compared to BL to Y1 in V group) or use baseline adjusted values when comparing end of study levels applies to all comparisons

We have now provided a comparison of the change in one group vs change in the other group (BL to Y1 in placebo, compared to BL to Y1 in V group) as suggested by the reviewer (new Fig. 1c and 1f). Of note, the results show again highly significant differences between the verapamil and the placebo group supporting our initial findings.

in fig 1M a correlation between chromogranin and AUC is provided, but it is not clear which patients are included, is it both baseline and end of study visits or only all at baseline or? (please clarify and only include each subject once or you need to adjust for dependency

Thank you for pointing this out. All available samples were included in this correlation and we have therefore now reanalyzed the data and adjusted for dependency using a repeated measures

correlation coefficient by mixed model (revised Fig. 1h). However, the correlation remained highly significant ($p=0.0026$).

given the very small numbers it would be relevant if possible to discuss if a larger clinical trial assessing this is ongoing/planned

A larger, independent clinical trial is in fact ongoing, and we have now added this information to the revised discussion (Ver-A-T1D, NCT04545151).

Reviewer #2 (Remarks to the Author):

Xu et al report on the impact of verapamil on type 1 diabetes through evaluation of clinical follow-up from previously reported clinical trial, samples from the trial, and in vitro studies of verapamil on islets. Key findings reported by the investigators were that verapamil decreased Tfh cells and IL21 and increased chromogranin A and that continuous use of verapamil stabilized beta cell function. Further, the investigators report that verapamil induced anti-oxidative and anti-apoptotic genes expression in human islets. The investigators conclude that the studies further support the benefits of verapamil and provide mechanistic insights.

Strengths: Novel observation re: changes in chromogranin A

Weaknesses:

Major: There are insufficient details regarding experimental and analysis methods and human subjects in this paper and methodologic issues that limit the ability to determine whether the author's conclusions are valid.

We appreciate Reviewer #2 recognizing the strengths, novelty, and mechanistic insight of the studies, acknowledge the criticism, and apologize for not having included more details regarding experimental and analysis methods and human subjects. We have now provided all this information, removed some of the unneeded figure panels pointed out by the reviewer as detailed below and believe that the revised and streamlined paper is greatly improved and now clearly supports our conclusions.

For the clinical results, there is no information about the study protocol for which data is presented about those in the active treatment arm who continued or did not continue to receive ongoing active study drug and those who were in the control arm at the conclusion of the primary trial. For the mechanistic results, there is no data as to whether the samples were masked before testing and analysis. Further, the text states that serum was available on 20 individuals, yet there are variable numbers of subjects with data for each of the figures. There is insufficient information about "healthy" and other T1D individuals used in some experiments/figures. There is no information about whether these are the same or different subjects for the different experiments.

We have now included more information in the revised text and revised/new tables about the study protocol and the 5 subjects in the active treatment arm who continued verapamil, the 4 subjects who did not continue to receive ongoing active study drug and the 6 participants who were in the control arm at the conclusion of the primary trial and never received any study drug (Supplemental Table S1 and S4). Importantly, treatment groups were well balanced in terms of baseline subject characteristics overall and in regard to the subset providing the 20 samples used for serum proteomics and PBMC qPCR (revised Supplemental Table S1), as well as in regard to the subject characteristics of the subgroups at the start of year 2 (new Supplemental Table S4). Also, experimenters were blinded to the study group allocation of the samples during mechanistic studies. In addition, we have also explained and provided more information about the healthy, non-diabetic subjects mentioned and have clarified that the same subjects were used for the different experiments (revised Supplemental Table S1 and text).

For the qPCR data, there is insufficient description of methods. The reference cited regarding qPCR methods refers back to another paper for methods in animal models from 2008. Data is expressed as relative expression values which is common, but there is no information about absolute levels of each transcript, and no information about how the relative expression assessments are made.

We have expanded the description for the qPCR methods. In brief, total RNA was extracted using the miRNeasy Mini Kit (Qiagen) according to the manufacturer's instructions. RNA was reverse transcribed to cDNA using the first strand cDNA synthesis kit (Roche) and quantitative real-time PCR was performed using a LightCycler 480 system (Roche) (Jo S et al. Endocrinology 2021) and the primers listed in Supplemental Table S6. Relative gene expression was determined using the $2^{-\Delta\Delta CT}$ method as previously described in detail (Livak KJ et al. Methods 2001). All results were normalized for GAPDH run as an internal standard. To further assure validity of the $2^{-\Delta\Delta CT}$ calculations, we also conducted serial template dilutions and confirmed comparable target and reference amplification efficiency.

It is challenging to understand why the investigators report global STAT4 expression rather than investigating phosphorylated stat. Similarly, QPCR in PBMC is an unusual way to measure IL21. Information about the limit of quantitation for serum IL21 should be included in the methods. It is unclear why flow or cytof was not used to evaluate CXCR3 or CXCR5 cells. It is unclear if the same three islet preparations were used for Figure 4 C and 4 D.

These are excellent points and the reason why the current tests were performed rather than others, is sample availability. In particular, we unfortunately had only a limited amount of PBMCs available allowing for qPCR, but not Western of phosphorylated STAT4. In addition, these PBMCs were not viable precluding flow or cytof to evaluate CXCR3 or CXCR5. While indeed not that common, there have been reports of successful IL21 measurements by qPCR using PBMCs and these references have been added (Kenefleck et al. JCI 2015, Publicover et al. JCI 2011, Xu et al. PLOS One 2013). In terms of serum IL21, we have also added the limit of quantitation (0.32-5000 pg/mL) to the methods. For original Figure 4c and 4d the same three islet preparation were used, and we therefore have eliminated Figure 4d as recommended below by the reviewer.

Other:

Figure 1A/B data is the same as shown in a different way without added value in Figure C/D. Thus, the latter are not needed. The same is true for Figure E/F data which is duplicated in Figures 1G/H.

As suggested by the reviewer, we have removed the original Fig. 1c/d and Fig. 1g/h.

Figure 1J shows a well-known feature of T1D, comparing C-peptide in healthy individuals and T1D (and no information as to who these samples are from), and does not add value to the manuscript.

We have also eliminated Fig. 1j as recommended.

Figure 1K/L represents data from a subset of individuals in the original clinical trial; thus, it is not clear what this figure is aiming to demonstrate.

We agree with the reviewer and have also removed original Fig. 1k/l.

Figure M has >25 data points. This suggests that these are multiple data points from the same individuals and thus other statistical approaches are needed.

Thank you for pointing this out. We have now taken a different statistical approach using a repeated measures correlation coefficient by mixed model and have reanalyzed the data (revised Fig. 1h). Of note, the correlation remained highly significant ($p=0.0026$).

Figure 2A has a limited number of individuals; data would be better shown with individual results. Figure 2C cannot be interpreted without more information about the protocol and the blinding and selection and distributions of subjects for the f/u trial. Figure 2D is redundant to 2C (which does not have information about the DISCV group). A box plot is inappropriate way to illustrate the HbA1c values in the two arms and it is also missing the DISCV group.

Some corresponding individual results are already shown in Figure 1d-e, but in order to provide a clear visual comparison of the time courses, we feel it is important to show the line graph of Figure 2a. As for Figure 2c, we have now provided a lot more information about the subjects for the f/u study in the revised text and especially the new Supplemental Table S4 provides a detailed break down of the distribution of these subjects. Also, as per the reviewer's suggestions, we have removed the original Figure 2d and have replaced the HbA1c box plot with the appropriate line graph providing the corresponding time courses and including the Disc V group (new Fig. 2d).

Figure 3: More information needed about precise methods used for RNA seq and qPCR studies and analysis. Why were only 5 people per group included in the qPCR analysis? Figure 3 E; the differences between control and verapamil treatment are dictated by 2 extreme values. Is there a relationship between serum IL21 levels and the relative RNA expression?

We have now included a lot more information about the precise methods used for RNAseq and qPCR in this revised manuscript. Also, the 5 people per group shown in the PBMC qPCR analysis are the same subjects that provided the serum for the proteomics analysis and for whom enough sample material (PBMCs and serum) was available at baseline and at year 1. This has now been clarified in the text and in the revised Supplemental Table S1. We appreciate the reviewer pointing out the 2 extreme values in Figure 3e. Indeed, reanalysis of these 2 outliers revealed still elevated, but a lot lower values for these samples. For consistency and to confirm that the problem with the previous run was isolated to these 2 samples, we also reanalyzed the other samples of this comparison, and this revised data is now shown in Fig. 3e. Of note, the difference remained significant. Also, we did find a relationship between serum IL21 levels and the relative RNA expression in PBMCs ($R=0.56$, $p=0.026$).

Figure 4: Information about methods for this experiment are lacking detail (dose and time of exposure to verapamil), impact of verapamil on non-islet tissue? How does the dose used in these in vitro experiments relate to the therapeutic dose in the clinical trials? If the same islets and experiment were used to generate the data, there is no added value to Figure 4C and 4D (i.e. measuring RNA by 2 approaches that would be expected to correlate)

We have now included more information about the methods for this experiment. Also, the advantage of this ex vivo experiment is that isolated handpicked islets were used, making any confounding effects from verapamil acting on non-islet tissue very unlikely. Verapamil doses used clinically and in vitro represent typically used doses. In particular, the concentration used in these in vitro experiments had been established in early work to study the effects of verapamil in beta cells and islets and we now provide these references. Since the same islets were used in Figure 4c and d, we have now removed the original Figure 4d following the reviewer's recommendation.

Text:

It seems odd and out of place to characterize MMTT as “cumbersome for patients and health care providers”.

This text has been stricken.

There are many well-established parameters for validation of biomarkers depending upon the intended use. While the CHGA data is intriguing, the author's statements about the utility of the currently reported measure as compared to selected other measures is not well supported.

We have revised the discussion of this aspect accordingly.

References 11-13 describe IL-21 production by CD4 T cells and/or association with Tfh cells. There is no reference to previous reports on serum IL-21 levels in T1D.

We have now added a reference on serum IL-21 levels in T1D (Baharlou R et al. Immunol Invest 2016).

For the reader's benefit, it would seem worthwhile to reference ongoing RCTs using verapamil in T1D.

Excellent point, those trials are now referenced in the discussion.

Reviewer #3 (Remarks to the Author):

The authors previously published a study about the beneficial effects of Verapamil on new onset type 1 diabetes patients. Interestingly, Verapamil is an approved and generic drug currently used for high blood pressure. In this paper, the authors perform plasma proteomics on a small cohort at base line and after one year. They also perform gene expression studies on islets with Verapamil as well.

In the plasma proteomics study, they find the most significantly changed protein to be chromogranin A (CHGA), which is known to be produced in beta cells and which has recently been implicated as an auto-antigen in T1D. The authors discuss the potential roles of CHGA in the context of T1D. In the gene expression study, they focus on down regulation of the thioredoxin system and how this might be beneficial to beta cells.

Overall, I find this study quite interesting and the findings potentially significant. My major reservations are the size of the cohort, which consists of only 9 T1D subjects given the drug, 6 control T1D and 6 healthy controls. The plasma proteomics pipeline was not easy to follow for this reviewer, mainly due to the use of custom scripts, but the large issue here is that the plasma proteomics results are nowhere documented at all, except for log2 changes of the top significant proteins. As the authors used a quite involved set up with TMT labeling, depletion, a reference channel etc. there is ample opportunity for variation. This is hard to judge given that no data is available. However, from this reviewer's experience, it would be hit or miss with such a small number of cases. CHGA goes down for 4 of the 9 subjects in Fig. 1b (where are the others?), but the controls go up or down as well, both in Fig. 1A and Fig. 1e. In conclusion, the plasma proteomics study is potentially interesting but not convincing as it is.

The gene expression study is relatively standard as far as I can tell and the observed effects quite broad. Altogether, the study lacks rigor that would allow it to carry the far reaching interpretations that the authors make.

We would like to thank Reviewer #3 for the overall interest in our study and for the detailed comments. We have addressed all of them as detailed below and doing so has not only increased the rigor and provided more evidence supporting our initial conclusions, but has also expanded the scope and clearly strengthened the paper.

In more detail:

p.1, title, line 1

- 'omics reveal ...'

- Omics analysis is not sufficient to fit the title ® see later explanations

We apologize for previously not having provided more information in regard to the omics analyses performed and have rectified this issue in this revised manuscript. Nevertheless, we have changed the title and eliminated the term 'omics' as suggested by the reviewer.

p.2, Abstract, line 3-4

- 'exact biological changes verapamil elicits in humans with T1D, how long they may last'

- Addressing only biological changes on specific picked proteins
 - o Proteomics: one single protein in serum proteome (CHGA)
 - o RNAseq: 5 proteins picked

- o Others: c-pep, T-cell markers (CD4, CXCR3, STAT4, CXCR5, IL21)
- Time scale of omic analysis: only 1 time point (1 year)
- o Where is the time point of 3 month if you compare it to your paper before (Ovalle et al.)
- o Why no serum proteomics on second year samples?
- Additionally, RNA seq with only emphasize on own parameters without telling more about the outliers

In this revised manuscript we have added a more global description of the verapamil associated biological changes in terms of serum proteomics (new Supplemental Table S3) as well as in terms of islet RNAseq (new Supplemental Table S5) and discuss the findings in the revised text.

We should further clarify that the proteomics study was performed using longitudinal samples from the same 10 subjects at baseline and at 1 year of placebo or verapamil treatment (Fig. 1a-b and new Supplemental Figure S1) and the time factor was included in the regression analysis of the data. Of note, this analysis already included 20 samples and unfortunately, additional analyses of 3 month or second year samples were not feasible primarily due to lack of sufficient serum samples. Nevertheless, we demonstrate changes over time for the top protein changed using ELISA which required less serum and allowed inclusion of second year samples (Fig. 2a).

In addition, we now also highlight and discuss the RNAseq ‘outliers’ of extreme up- and downregulation (revised Fig. 4a-c and text).

p.2, Abstract, line 3-4

- ‘unbiased proteomics analysis’

- Unbiased proteomic analysis completely missing
- Seems like a targeted proteomics analysis in the main text description

Based on the reviewer’s comments, we have made major revisions to the manuscript and now provide a more complete explanation and demonstration of our global proteomics analysis (new Supplemental Figures S1 and S2, Supplemental Table S3) and have revised the text accordingly.

p.2, Abstract, line 6

- ‘effects of continuous verapamil use over a 2-year period’

- Only shown c-pep AUC and CHGA level (measured by ELISA?) for second year point

We actually demonstrate the time course for 4 clinically important and relevant parameters including C-peptide AUC, CHGA, insulin requirements and HbA1C and have clarified this in the text and revised Fig. 2.

p.2, Abstract, line 10-11

- ‘Our present results reveal that verapamil regulates the thioredoxin system and promotes anti-oxidative and anti-apoptotic gene expression profile in human islets, ... ’

- Those results were already known and can’t be claimed as novel in this abstract (Ovalle et al.; Xu et al.; Chen et al.)

To clarify, the present work represents the first gene expression profile in human islets in response to verapamil, nevertheless we have changed the wording in the abstract to ‘..., we confirmed by RNA-sequencing analysis that verapamil regulates the thioredoxin system and promotes an anti-oxidative and anti-apoptotic gene expression profile...’ to avoid any potential misunderstanding.

p.3, results, line 1-2

- ‘treatment groups were well balanced in terms of subject characteristics’
- Healthy group only has female participants and a BMI above 27 (vs. 24)
- Describe this gender bias in main text
- If so compare only females in T1D vs. females of healthy

The healthy group did not receive any treatment, was not part of the clinical trial and samples were only used as a reference and comparison for some of the exploratory studies. Nevertheless, the reviewer’s point about gender bias was well taken and we have therefore now included additional male samples in this group to provide a better comparison. We now have clarified these points in the text and the revised Supplemental Table S1 and have updated the data accordingly. Of note, they confirmed our original results. (Since subjects with T1D inherently have lower BMIs, the slightly higher BMI of now 26.4 in the healthy subjects cannot be avoided.)

p.3/4, results, line 21 & 1-2

- ‘unbiased proteomics analysis’ & ‘53 proteins or peptides whose relative abundance over time was significantly altered ...’
- Unbiased proteomic analysis completely missing
- ® need to provide introduction figure about proteomic data set like study design, measured protein numbers, CV values, ...
- Experimental description of proteomic dataset totally missing in main text and method description not sufficient
- No comment to any of the other significant proteins/peptides
- ® description about picking CHGA insufficient and looks like a targeted approach not an unbiased analysis
- If you want to address the question of exact biological changes you need to use an unbiased analysis and comment on other significant proteins
- How were people for serum proteomic analysis picked (only 5 out of 6/9) ® no description and explanation provided
- Serum proteomics analysis of healthy people totally missing ® gender bias?

In this revised manuscript, we have now provided a more detailed description of the proteomics analysis including an introduction figure about the proteomics data set as well as a detailed experimental and methods description demonstrating the data quality (Supplemental Figure S1 and S2). Of note, the protein quantification was overall robust with most of the proteins quantified from both batches of TMT experiments with all 20 data points and in fact, only proteins with TMT reporter ion intensity data in all channels from each batch experiment were considered quantified and the numbers are now provided (Supplemental Figure S2).

We have now also included a discussion of a number of additional proteins that were significantly changed in the revised text as well as the rationale for focusing on CHGA. In addition, we provide a gene ontology analysis for biological process enrichment based on all proteins that were significantly changed and the results further support our overall conclusions (new Supplemental Table S3).

Proteomics samples were selected based on availability of usable serum (quantity and quality) and we have now clarified this point. We have also clarified that healthy people did not receive any treatment, were not part of the clinical trial and as such no longitudinal serum samples were available for any proteomic analysis. On the other hand, we have rectified the potential gender bias raised by the reviewer as mentioned above by now including female and male samples in this group.

p.5, results, line 1-2

- Fig 1 J-L: is verapamil BL vs. Y1 significant? Is healthy vs. verapamil BL/Y1 significant?

To comply with the request of reviewer 2, Fig. 1J-L has been eliminated. However, while the C-peptide AUC of the verapamil group at BL was significantly lower as compared to healthy ($P < 0.001$) that difference was diminished during the 1 year of verapamil treatment and no longer significant at Y1 vs healthy.

p.5, results, line 4

- 'a highly significant inverse correlation'
- too strong wording at a correlation of -0.603

The wording has been softened and the term 'highly' deleted.

p.6, results, line 21

- 'interleukin 21 (IL21) were elevated >10-fold or 100-fold ...'
- differences of IL21 only because of 2 outlier patients
- ® needs to be discussed and may need additional replicates to claim this topic

We appreciate the reviewer alerting us to these 2 outliers. Indeed, reanalysis revealed still elevated, but a lot lower values for IL21. For consistency and to confirm that the problem with the previous run was isolated to these 2 samples, we also reanalyzed the other samples of this comparison and this revised data is now shown in Fig. 3e. Of note, while the observed elevation was lower, it remained significant ($P = 0.03$).

p.7, results, line 14

- 'this genomic analysis'
- RNA seq is transcriptomic analysis
- Global transcriptomic analysis again missing and no description of data set: significant transcripts, outliers, ...
- Introduction of RNAseq data set needed ® no description in main text
- Fig 4b: enrichment analysis needs to be displayed globally to address "exact biological changes"
- o This doesn't present an omic analysis

- Best would have been if matched experiment that omic analysis could have been compared to serum proteomics ® how is the correlation?

We corrected the designation of RNAseq as transcriptomic analysis.

In this revision, we have also added an introduction and much more complete description of the RNAseq data set with demonstration and discussion of significantly changed transcripts including the most extreme ones in the revised main text and in the revised Fig. 4a-c.

We have also added a global enrichment analysis (Enrichr) shown in new Supplemental Table S5.

While obviously theoretically best from a scientific standpoint, a matched experiment of serum proteomics and islet transcriptomics is not possible in this case of a verapamil study in living human subjects as isolated islets are only obtained from brain dead organ donors. Also, given the difference in samples and the different proteins and genes found in circulation and in islet tissue, no correlation at the single gene level would be anticipated. However, based on the reviewer's helpful comments and addition of global enrichment analysis for both serum proteomics and serum transcriptomics, we were able to identify a striking correlation related to the enrichment of three neutrophil-mediated processes (immunity, degranulation, and activation) in response to verapamil. This further supports our initial conclusion that verapamil may also have immunomodulatory effects and this interesting finding further strengthens our revised paper.

p.7, results, line 15

- 'was up- as well as downregulated'

- Specify numbers and discuss outliers a little
- cherry picking on those which were already know (low p-values, low log2 fold change)
- Does not reflect the aim of discussing exact biological changes and of an omic analysis

We have now specified the numbers of up-and downregulated genes and discuss the 'outliers' and the genes with highly significant, high log2 fold changes. As mentioned above, we have also revised Fig. 4a-c accordingly and have added a global enrichment analysis (new Supplemental Table S5).

p.10/11, methods, Proteomics /

- 'desalted by solid-phase extraction (SPE)'

- What kind of SPE?

- 'one pooled sample was generated by pooling an aliquot of peptides from each sample to serve as a "universal reference"

- Description not sufficient: how was sample pooled, equally? Amount of peptides?
- Description of referencing missing

- 'the TMT-labeled peptides combined from all 11 channels ...'

- Which samples were combined in one TMT experiment?
- Description not enough

- 'data processing and analysis, ... mzRefinery ... MS-GF+ ...'

- MS raw data needs to be provided on PRIDE
- Data processing description not sufficient

- Using of a not standard software[®] test data processing in gold standard software, e.g. MaxQuant
- Provide all parameters set in software, 1% FDR and PepQValue < 0.005 not sufficient: e.g. ppm limits, peptide length parameters, ions, ...
- Describe exactly filtering and post processing of dataset

All the information is now provided in the revised method section along with the new workflow Supplemental Figure S1. We should also note that the serum proteomics workflow we applied is a relatively standardized workflow implemented for large-scale biomarker studies. As recently described by Keshishian et al. (Nature Protocols 2017, <https://doi.org/10.1038/nprot.2017.054>), the overall workflow with depletion, TMT/itraq labeling, and fractionation is highly reproducible with CV across process replicate to be <12%. Our laboratory observed similar levels of reproducibility.

The MS raw datasets can be found in the online repositories: Massive.ucsd.edu with accession: MSV000087598. (For now, manuscript reviewers can access the data via FTP using password: Vera3545). The data will also be available through ProteomeXchange with accession: PXD026601.

p.11, methods, Genomics / RNA Sequencing

- 'Genomics / RNA sequencing'

- Dataset needs to be provided for reviewers
- RNA sequencing provides no information about genomics
- Data set description not sufficient
- FDR model not mentioned and described

The RNAseq dataset is now available for reviewers at GEO with accession # GSE181328 and token: udiricaefdkvhup

As mentioned above, we have corrected the wording and eliminated the term 'genomics'.

We have also added a more complete description of the data set in the revised manuscript.

The FDR model is now mentioned and described in the Methods section.

p.15/16, Figure legends

- description of figure legends not sufficient to understand the figures[®] need to describe everything in more detail
- labels of x axis and y axis not sufficient[®] especially a-d: CHGA (log2) / CHGA: are those log2 values on raw values? Relative values? Needs a more precise description
- 2a How measured are CHGA values measured? by ELISA?
- 4: no description of p-value adjusted

We have provided a more detailed description of the figure legends. In particular, we have clarified that the y-axis in Figure 1 (a-d) is relative abundance values in zero-centered log2 form. In addition, we have also clarified that the CHGA values in Fig. 2a were measured by ELISA. Moreover, for Fig. 4 we now provide a description of the adjusted p-value.

REVIEWER COMMENTS

Reviewer #1 (Remarks to the Author):

Thanks to the authors for responding to my questions, no further questions

Reviewer #2 (Remarks to the Author):

Xu et al have revised their report on the impact of verapamil on type 1 diabetes. They have addressed several of the specific questions raised previously as well as adding new information.

The key strength of the paper is the novel observation regarding changes in chromogranin A in people treated with verapamil.

While this as well as some of the other data presented is intriguing; the limitations of the study due to subject and sample availability necessitate caution in interpretation of the results which should be considered as interesting and provocative preliminary data. Thus, many statements in the paper should reflect this.

Specific comments:

Title: While the authors enthusiasm is evident, the title of the manuscript should be more reflective of the preliminary nature of the observations.

Summary and Discussion: The caveats of the small, self-selected N, open label treatment on clinical variables such as Hba1c and insulin use should temper the discussion.

The authors have inserted a bit more information about subject and sample collection, but it remains difficult for the reader to understand. The suggestions below may be helpful.

- As the authors are reporting clinical outcome data as well as mechanistic data stemming from a clinical trial, a consort diagram would be a useful addition to the paper starting with the N randomized to the treatment and placebo groups in the initial trial, the N in each group at the outcome of the initial trial (1 year), and then the outcome of the f/u after that time. This should make clear the number of subjects followed for 2 years and the number of these that had samples for the experiments.
- It appears that there are not more than 20 individual's samples/data used in this study. Thus, a comprehensive table with subject number, treatment group, key relevant information such as age, gender, race, BMI, HbA1c at baseline, 1 and 2 years, insulin use at baseline 1 and 2 years, MMTT data at baseline 1 and 2 years, and whether their sample was used in what experiment / figure would greatly facilitate the reader's interpretation of results.

The manuscript indicates that verapamil "reverses T1D-induced elevations in circulating proinflammatory T-follicular helper cells" in the abstract, results, and discussion. However, this statement is not well supported since the authors did not have viable PBMC to test this question by flow cytometry. I am sympathetic regarding the absence of viable samples. However, the qPCR data were performed on total PBMC, which contains a mix of cell populations among which Tfh are relatively rare – thus it is problematic to ascribe the transcript levels solely to Tfh. For example, CXCR5 (the main Tfh marker used here) is also expressed on B cells and some subsets of dendritic cells.

The Enrichr/GO term data is of interest and adds to the manuscript. However, the authors should provide the number of genes in each term and the number of genes enriched in their dataset for both Supplemental Tables 3 and 5. This would provide important context for evaluating the statements on pages 4 and 8. In particular, understanding the neutrophil signature described on page 8 is challenging given that the signature is being detected in islet preps. Are the genes detected in the neutrophil pathway overlapping with those used by alpha or beta cells in protein secretion (and thus this is not a "neutrophil" signature per se)? Do the authors have an alternative explanation to explain

neutrophil-specific gene up/down regulation in islet preps?

Figure 1: The authors state that Figure 1h shows the utility of serum CHGA as a biomarker of disease progression. However, what is shown in the figure is a correlation between CHGA and C-peptide. The authors have longitudinal data on a small number of individuals, what is the relationship between change in C-peptide and change in CHGA for each individual and did this relationship vary according to treatment group?

Figure 2: This figure should show individual data plots, or if too busy in the main figure, a supplemental figure should show the data in that way.

Figure 3: Since the point of Figure 3 is to show that verapamil reduced CXCR5 and IL21 expression as well as serum IL21, it would be more informative to link the individual data like shown in figure 1 a and b. Note: In response to previous query regarding serum IL21 measurements, the authors note a reference from Baharlou, 2016. Yet the values from healthy non T1D in that paper of serum IL21 was around 180 pg/ml while the values from healthy non T1D in this manuscript appear to be less than 50 pg/ml. Please comment.

Figure 4C could be better displayed by showing the read count data for each gene before and after verapamil. This would be particularly helpful in shedding light on the changes in HLA A-B-C genes. The current heat map seems to indicate that only one of 3 replicate islet preps downregulated these genes, while the other two did not. Thus, it is difficult to understand whether the data supports the authors conclusion about the effect of verapamil on these genes. Better understanding the variability between preps by presentation of read counts (or normalized read counts +1) would aide in interpretation of this data.

Page 6: It is challenging to understand how a variable with a correlation with an r^2 of <0.4 to MMTT C-peptide is a suitable biomarker of disease progression. Please comment.

Page ¾ (minor): Since these subjects were not in the groups due to randomization, but rather self-selected, consider changing the sentence with the phrase “well balanced” to something such as “there were no significant differences between these groups with respect to”

Page 8 (minor): INSIG1 and GP2 are indicated as shown in Fig 4C but are not present in the heatmap.

Page 8: (minor) Consider re-wording description of islet experiments in main text for clarity. The methods state that islets from 3 donors were used and these islets were treated with or without verapamil and then used in the experiments. The main text talks about verapamil or untreated samples from 3 individuals which may confuse the reader to think that the experiment involved added serum from the trial participants to islets.

Other: The authors indicate that the proteomics and transcriptomics data have been deposited in appropriate publicly available databases. The authors should also make the other data supporting this manuscript available including C-peptide, insulin dose, etc.

Reviewer #3 (Remarks to the Author):

The authors have done a credible revision and this reviewer now agrees to publication

REVIEWER COMMENTS

Reviewer #1 (Remarks to the Author):

Thanks to the authors for responding to my questions, no further questions

Thank you!

Reviewer #2 (Remarks to the Author):

Xu et al have revised their report on the impact of verapamil on type 1 diabetes. They have addressed several of the specific questions raised previously as well as adding new information.

The key strength of the paper is the novel observation regarding changes in chromogranin A in people treated with verapamil.

While this as well as some of the other data presented is intriguing; the limitations of the study due to subject and sample availability necessitate caution in interpretation of the results which should be considered as interesting and provocative preliminary data. Thus, many statements in the paper should reflect this.

The text has been revised accordingly.

Specific comments:

Title: While the authors enthusiasm is evident, the title of the manuscript should be more reflective of the preliminary nature of the observations.

We have revised the title accordingly. ("Exploratory Study Reveals Far Reaching Systemic and Cellular Effects of Verapamil Treatment in Subjects with Type 1 Diabetes").

Summary and Discussion: The caveats of the small, self-selected N, open label treatment on clinical variables such as Hba1c and insulin use should temper the discussion.

We have tempered the discussion taking into account these caveats.

The authors have inserted a bit more information about subject and sample collection, but it remains difficult for the reader to understand. The suggestions below may be helpful.

- As the authors are reporting clinical outcome data as well as mechanistic data stemming from a clinical trial, a consort diagram would be a useful addition to the paper starting with the N randomized to the treatment and placebo groups in the initial trial, the N in each group at the outcome of the initial trial (1 year), and then the outcome of the f/u after that time. This should make clear the number of subjects followed for 2 years and the number of these that had samples for the experiments.
- It appears that there are not more than 20 individual's samples/data used in this study. Thus, a comprehensive table with subject number, treatment group, key relevant information such as age, gender, race, BMI, HbA1c at baseline, 1 and 2 years, insulin use at baseline 1 and 2 years, MMTT data at

baseline 1 and 2 years, and whether their sample was used in what experiment / figure would greatly facilitate the reader's interpretation of results.

We appreciate the Reviewer's suggestions and now have added a reference to the published consort table of the initial study as well as a comprehensive table including subject number and all HIPAA-allowable clinical data, including treatment group, BMI, HbA1c at baseline, 1 and 2 years, insulin use at baseline 1 and 2 years, MMTT-stimulated C-peptide AUC data at baseline 1 and 2 years, and what experiment / figure their sample was used in (new Supplemental Table S6). (All demographic data are summarized in Supplemental Tables S1 and S4.). Together, this should help any reader get a good understanding of the subjects and samples used in this study.

The manuscript indicates that verapamil "reverses T1D-induced elevations in circulating proinflammatory T-follicular helper cells" in the abstract, results, and discussion. However, this statement is not well supported since the authors did not have viable PBMC to test this question by flow cytometry. I am sympathetic regarding the absence of viable samples. However, the qPCR data were performed on total PBMC, which contains a mix of cell populations among which Tfh are relatively rare – thus it is problematic to ascribe the transcript levels solely to Tfh. For example, CXCR5 (the main Tfh marker used here) is also expressed on B cells and some subsets of dendritic cells.

We have rephrased this statement in the abstract, results and discussion to be more accurate.

The Enrichr/GO term data is of interest and adds to the manuscript. However, the authors should provide the number of genes in each term and the number of genes enriched in their dataset for both Supplemental Tables 3 and 5. This would provide important context for evaluating the statements on pages 4 and 8. In particular, understanding the neutrophil signature described on page 8 is challenging given that the signature is being detected in islet preps. Are the genes detected in the neutrophil pathway overlapping with those used by alpha or beta cells in protein secretion (and thus this is not a "neutrophil" signature per se)? Do the authors have an alternative explanation to explain neutrophil-specific gene up/down regulation in islet preps?

We appreciate the Reviewer recognizing the added value provided by the Enrichr/GO term data and now also provide the number of genes enriched in our dataset and the number of genes in each term (overlap) as requested (revised Supplemental Tables S3 and S5). In regard to the neutrophil signature found in the islets, we added these data in response to reviewer #3's question and were initially surprised by it as well. However, while the large number of genes under these terms indeed contains genes involved in protein secretion that are also shared by alpha and beta cells, enriched genes in our dataset rather included genes associated with innate immunity such as complement component 3 (C3), inflammatory cytokines (macrophage migration inhibitory factor, MIF) and leucocyte cell surface molecules (cluster of differentiation, CD31, CD66c, CD93). Interestingly, more recently, the notion of pancreas-resident/infiltrating neutrophils has been established (*Citro A et al. Front Endocrinol 2021 Feb 8;11:606332*) and CD93 and CD31 have previously been found to be expressed in isolated pancreatic islets and islet endothelial cells (*Strawbridge RJ et al. Diabetes 2016 65: 2888-99; Menger MM et al. Acta Diabetol 2021 July 12*). We therefore speculate that the observed signature was at least in part due to changes in these islet-resident cells and have now added these considerations to our discussion of the data.

Figure 1: The authors state that Figure 1h shows the utility of serum CHGA as a biomarker of disease progression. However, what is shown in the figure is a correlation between CHGA and C-peptide. The authors have longitudinal data on a small number of individuals, what is the relationship between change in C-peptide and change in CHGA for each individual and did this relationship vary according to treatment group?

We have now included a demonstration of the relationship between change in C-peptide AUC and change in CHGA for each individual showing a clear and significant inverse correlation (new Supplemental Figure S3), further supporting the notion of CHGA as a potential biomarker of disease progression. (A meaningful comparison of any additional correlation broken down by treatment group does not seem to be possible due to the small sample size.)

Figure 2: This figure should show individual data plots, or if too busy in the main figure, a supplemental figure should show the data in that way.

We have added a supplemental figure including the individual data (new Supplemental Figure S4).

Figure 3: Since the point of Figure 3 is to show that verapamil reduced CXCR5 and IL21 expression as well as serum IL21, it would be more informative to link the individual data like shown in figure 1 a and b. Note: In response to previous query regarding serum IL21 measurements, the authors note a reference from Baharlou, 2016. Yet the values from healthy non T1D in that paper of serum IL21 was around 180 pg/ml while the values from healthy non T1D in this manuscript appear to be less than 50 pg/ml. Please comment.

The major goal of Figure 3 was to show the increase in CXCR5 and IL21 expression and serum IL21 in T1D as compared to non-diabetic controls and demonstrate the prevention or normalization of this pathology with verapamil. However, to further highlight the effects of verapamil as opposed to placebo on CXCR5 and IL21 as suggested by the Reviewer, we have now also linked the individual data like shown in figure 1a and b (new Supplemental Figure S5).

Serum IL21 measurements in healthy, non-diabetic adults have in general been reported to be at ~50pg/mL (*He Z et al. Br J Dermatol 2012 Jul;167:191-3; Mizutani H et al. Allergol Int 2017 Jul;66:440-444*) consistent with our results and we have added these references. (While we obviously don't know why the values in the Baharlou paper were higher, one possible explanation is that all their control subjects were under the age of 21.)

Figure 4C could be better displayed by showing the read count data for each gene before and after verapamil. This would be particularly helpful in shedding light on the changes in HLA A-B-C genes. The current heat map seems to indicate that only one of 3 replicate islet preps downregulated these genes, while the other two did not. Thus, it is difficult to understand whether the data supports the authors conclusion about the effect of verapamil on these genes. Better understanding the variability between preps by presentation of read counts (or normalized read counts +1) would aide in interpretation of this data.

We now have included a display of the normalized read counts for each gene before and after verapamil for all 3 individual islet donors (new Figure 4d). While the counts show the expected inter-

individual variability, they also demonstrate that the HLA A-B-C genes were downregulated in every individual.

Page 6: It is challenging to understand how a variable with a correlation with an r^2 of <0.4 to MMTT C-peptide is a suitable biomarker of disease progression. Please comment.

From a statistical standpoint, a correlation of $r>0.5$ or $r^2>0.26$ is actually considered strong (Cohen J. *A power primer. Psychological Bulletin 1992, 112: 155–159*) and as such supports CHGA as a potential biomarker. Of note though, the suggestion of CHGA as a potential biomarker of disease progression is based on several additional observations aside from its correlation with an established biomarker (Fig. 1h and new Supplemental Figure S3c), including increase in diseased vs healthy individuals (Fig. 1g), longitudinal decrease with disease treatment, and increase in response to treatment discontinuation (Fig 2a). Obviously, the sensitivity and specificity of CHGA as a biomarker will have to be validated in further studies with large sample size and we have clarified these points in the revised discussion.

Page ¾ (minor): Since these subjects were not in the groups due to randomization, but rather self-selected, consider changing the sentence with the phrase “well balanced” to something such as “there were no significant differences between these groups with respect to”

This sentence has been changed as suggested.

Page 8 (minor): INSIG1 and GP2 are indicated as shown in Fig 4C but are not present in the heatmap.

These genes were considered ‘outliers’ by Reviewer #3 and therefore were not included in the heatmap. However, they are now included in the display of the normalized read counts (new Fig. 4d).

Page 8: (minor) Consider re-wording description of islet experiments in main text for clarity. The methods state that islets from 3 donors were used and these islets were treated with or without verapamil and then used in the experiments. The main text talks about verapamil or untreated samples from 3 individuals which may confuse the reader to think that the experiment involved added serum from the trial participants to islets.

The main text has been re-worded accordingly.

Other: The authors indicate that the proteomics and transcriptomics data have been deposited in appropriate publicly available databases. The authors should also make the other data supporting this manuscript available including C-peptide, insulin dose, etc.

These data have now been made available and are included in Supplemental Table S6.

Reviewer #3 (Remarks to the Author):

The authors have done a credible revision and this reviewer now agrees to publication

Thank you!

REVIEWER COMMENTS

Reviewer #2 (Remarks to the Author):

The authors have made many of the requested changes.